# Robust Robotic Control from Pixels using Contrastive Recurrent State-Space Models

## Abstract

Modeling the world can benefit robot learning by providing a rich training signal for shaping an agent's latent state space. However, learning world models in unconstrained environments over high-dimensional observation spaces such as images is challenging. One source of difficulty is the presence of irrelevant but hard-to-model background distractions, and unimportant visual details of task-relevant entities. We address this issue by learning a recurrent latent dynamics model which contrastively predicts the next observation. This simple model leads to surprisingly robust robotic control even with simultaneous camera, background, and color distractions. We outperform alternatives such as bisimulation methods which impose state-similarity measures derived from divergence in future reward or future optimal actions. We obtain state-of-the-art results on the Distracting Control Suite, a challenging benchmark for pixel-based robotic control.

## 1 Introduction

For a robot, predicting the future conditioned on its actions is a rich source of information about itself and the world that it lives in. The gradient-rich training signal from future prediction can be used to shape a robot's internal latent representation of the world. These models can be used to generate imagined interactions, facilitate model-predictive control, and dynamically attend to mispredicted events. Video prediction models (Finn et al., 2016) have been shown to be effective for planning robot actions. Model-based Reinforcement Learning (MBRL) methods such as Dreamer (Hafner et al., 2020) and SLAC (Lee et al., 2020) have been shown to reduce the sample complexity of learning control tasks from pixel-based observations.

These methods learn an action-conditioned model of the observations received by the robot. This works well when observations are clean, backgrounds are stationary, and only task-relevant information is contained in the observations. However, in a real-world unconstrained environment, observations are likely to include extraneous information that is irrelevant to the task (see Figure 1a). Extraneous information can include independently moving objects, changes in lighting, colors, as well as changes due to camera motion. High-frequency details, even in task-relevant entities, such as the texture of a door or the shininess of a door handle that a robot is trying to open are irrelevant. Modeling the observations at a pixel-level requires spending capacity to explain all of these attributes, which is inefficient.

The prevalence of extraneous information poses an immediate challenge to reconstruction based methods, e.g. Dreamer. One promising approach is to impose a metric on the latent state space which groups together states that are indistinguishable with respect to future reward sequences (DBC, Zhang et al. (2021)) or future action sequences (PSE, Agarwal et al. (2021)) when running the optimal policy. However, using rewards alone for grounding is sample inefficient, especially in tasks with sparse rewards. Similarly, actions may not provide enough training signal, especially if they are low-dimensional. On the other hand, observations are usually high-dimensional and carry more bits of information. One way to make use of this signal while avoiding pixel-level reconstruction is to maximize the mutual information between the observation and its learned representation (Hjelm et al., 2019). This objective can be approximated using contrastive learning models where the task is to predict a learned representation which matches the encoding of the true future observation (positive pair), but does not match the encoding of other observations (negative pairs). However, the

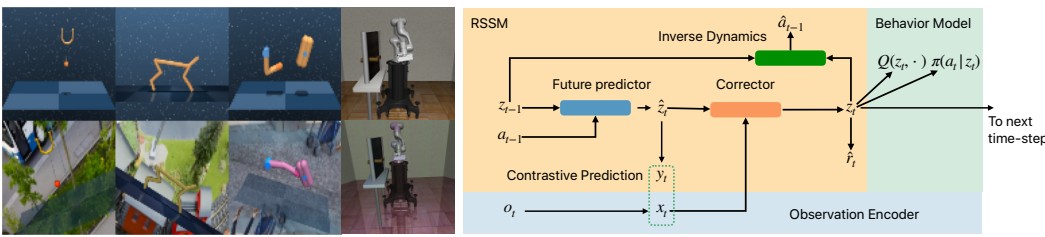

(a) Environment Examples  (b) Recurrent Contrastive State-Space Model

Figure 1: **Left**: Examples of visual control tasks in canonical (top) and distracting (bottom) environments. Background, color, and camera pose distractions significantly increase observation complexity. **Right** Our proposed **Co**ntrastive **Re**current State-Space Model (CoRe).

performance of contrastive variants of Dreamer and DBC has been shown to be inferior, for example, Fig. 8 in Hafner et al. (2020) and Fig. 3 in Zhang et al. (2021).

In this work, we show that contrastive learning can in fact lead to surprisingly strong robustness to severe distractions, provided that a *recurrent* state-space model is used. We call our model CoRe: **Co**ntrastive **Re**current State-Space Model. The key intuition is that recurrent models such as GRUs and LSTMs maintain temporal smoothness in their hidden state because they propagate their state using gating and incremental modifications. When used along with contrastive learning, this smoothness ensures the presence of informative hard negatives in the same mini-batch of training. Using CoRe, we get state-of-the-art results on the challenging Distracting Control Suite benchmark (Stone et al., 2021) which includes background, camera, and color distractions applied simultaneously. Video results from our model can be seen at supplementary/index.html.

## 2 MODEL DESCRIPTION

We formulate the robotics control problem from visual observations as a discrete-time partially observable Markov decision process (POMDP). At any time step $t$, the robot agent has an internal state denoted by $\boldsymbol{z}_t$. It receives an observation $\boldsymbol{o}_t$, takes action $\boldsymbol{a}_t$, and moves to state $\boldsymbol{z}_{t+1}$, receiving a reward $r_{t+1}$. We present our model with continuous-valued states, actions, and observations. A straightforward extension can be made to discrete states. The objective is to find a policy that maximizes the expected sum of future rewards $\sum_{t=0}^{\infty} \gamma^t r_t$, where $\gamma$ is a discount factor.

### 2.1 MODEL COMPONENTS

As shown in Figure 1b, our model consists of three components: a recurrent state space model (RSSM) (Hafner et al., 2019) which encodes the learned dynamics, a behavior model based on Soft-Actor-Critic (SAC) (Haarnoja et al., 2018) which is used to control the robot, and an observation encoder.

**Observation encoder**: This component is a convolutional network $f$ which takes an observation as input and outputs its encoding $\boldsymbol{x}_t = f(\boldsymbol{o}_t)$. LayerNorm (Ba et al., 2016) is applied in the last layer of the encoder.

**Recurrent State Space Model (RSSM)**: This component consists of the following modules,

$$
\begin{aligned}
\text{Future predictor} \quad & p(\hat{\boldsymbol{z}}_t | \boldsymbol{z}_{t-1}, \boldsymbol{a}_{t-1}), \\
\text{Observation representation decoder} \quad & \boldsymbol{y}_t = g(\hat{\boldsymbol{z}}_t), \\
\text{Correction Model} \quad & q(\boldsymbol{z}_t | \hat{\boldsymbol{z}}_t, \boldsymbol{x}_t), \\
\text{Inverse Dynamics model} \quad & \hat{\boldsymbol{a}}_{t-1} = a(\boldsymbol{z}_t, \boldsymbol{z}_{t-1}), \\
\text{Reward predictor} \quad & \hat{r}_t = r(\boldsymbol{z}_t).
\end{aligned}
$$

Each module is parameterized as a fully-connected MLP, except the future predictor which uses GRU cells (Cho et al., 2014). The future predictor outputs a prior distribution $\hat{\boldsymbol{z}}_t$ over the next state, given the previous state and action. This is decoded into $\boldsymbol{y}_t$ which corresponds to an encoding of the observation that the agent expects to see next. The correction model $q$ is given access to the current observation encoding $\boldsymbol{x}_t$, along with the prior $\hat{\boldsymbol{z}}_t$ and outputs the posterior distribution $\boldsymbol{z}_t$,

i.e., it corrects the prior belief given the new observation. The posterior state $\boldsymbol{z}_t$ is used to predict the reward, and also feeds into the actor and critic models that control the agent's behavior. The inverse dynamics model predicts the action that took the agent from state $\boldsymbol{z}_{t-1}$ to $\boldsymbol{z}_t$. The RSSM is similar to the one used in Dreamer and SLAC except that it includes an inverse dynamics model and that the observation representation is predicted from the *prior* latent state instead of the posterior latent state. These differences lead to slightly better results and a different probabilistic interpretation of the underlying model which we discuss in Section 2.2. Similar to Dreamer, the latent state $\boldsymbol{z}_t$ consists of a deterministic and stochastic component (More details in Appendix A).

**Behavior Model**: This component consists of an actor $\pi_\phi(a_t|z_t)$ and two critic networks $Q_{\theta_i}(z_t, a)$, $i \in \{1, 2\}$ which are standard for training using the SAC algorithm (Haarnoja et al., 2018).

## 2.2 Representation Learning with CoRe

Our goal is to represent the agent's latent state by extracting task-relevant information from the observations, while ignoring the distractions. We formulate this representation learning problem as dynamics-regularized mutual information maximization between the observations $\boldsymbol{o}_t$ and the model's prediction of the observation's encoding $\boldsymbol{y}_t$. Mutual information maximization (Hjelm et al., 2019) has been used extensively for representation learning. In our case, it can be approximated using a contrastive learning objective defined over a mini-batch of sequences, in a way similar to Contrastive Predictive Coding (CPC) (van den Oord et al., 2018). At time-step $t$ in example $i$ in the mini-batch, let $\boldsymbol{x}_{i,t}$ and $\boldsymbol{y}_{i,t}$ denote the real observation's encoding and the predicted observation encoding, respectively. The loss function is

$$L_c = -\sum_{i,t} \log \left( \frac{\exp(\lambda \boldsymbol{x}_{i,t}^\top \boldsymbol{y}_{i,t})}{\sum_{i',t'} \exp(\lambda \boldsymbol{x}_{i,t}^\top \boldsymbol{y}_{i',t'})} \right) - \sum_{i,t} \log \left( \frac{\exp(\lambda \boldsymbol{x}_{i,t}^\top \boldsymbol{y}_{i,t})}{\sum_{i',t'} \exp(\lambda \boldsymbol{x}_{i',t'}^\top \boldsymbol{y}_{i,t})} \right), \tag{1}$$

where $\lambda$ is a learned inverse temperature parameter and $i'$ and $t'$ index over all sequences and time-steps in the mini-batch, respectively. Intuitively, the loss function makes the predicted observation encoding match the corresponding real observation's encoding, but not match other real observation encodings, and vice-versa. In addition, the predicted and corrected latent state distributions should match, which can be formulated as minimizing a KL-divergence term

$$L_{\text{KL}} = \text{KL} \left( q(\boldsymbol{z}_t|\hat{\boldsymbol{z}}_t, \boldsymbol{x}_t) || p(\hat{\boldsymbol{z}}_t|\boldsymbol{z}_{t-1}, \boldsymbol{a}_{t-1}) \right). \tag{2}$$

An additional source of training signal is the reward prediction loss, $(r_t - \hat{r}_t)^2$. Furthermore, modeling the inverse dynamics, i.e. predicting $\boldsymbol{a}_t$ from $\boldsymbol{z}_t$ and $\boldsymbol{z}_{t+1}$ is a different way of representing the transition dynamics. This does not provide any additional information because we already model forward dynamics. However, we found that it helps in making faster training progress. We incorporate this by adding an action prediction loss. The combined loss is

$$J_M(\Theta) = L_c + \beta L_{\text{KL}} + \alpha_r (r_t - \hat{r}_t)^2 + \alpha_a ||\boldsymbol{a}_t - \hat{\boldsymbol{a}}_t||^2,$$

where $\beta, \alpha_r, \alpha_a$ are tunable hyperparameters and $\Theta$ is the combined set of model parameters which includes parameters of the observation encoder, the RSSM, and the inverse temperature $\lambda$.

**Relationship with Action-Conditioned Sequential VAEs** The above objective function bears resemblance to the ELBO from a Conditional Sequential VAE which is used as the underlying probabilistic model in Dreamer and SLAC. The objective function there is the sum of an observation reconstruction term $||\boldsymbol{o}_t - \hat{\boldsymbol{o}}_t||^2$ and $L_{\text{KL}}$ scaled by $\beta$, corresponding to a $\beta$-VAE formulation (Higgins et al., 2017). This formulation relies on decoding the *posterior* latent state to compute $\hat{\boldsymbol{o}}_t$. Since the posterior latent state distribution $q(\boldsymbol{z}_t|\hat{\boldsymbol{z}}_t, \boldsymbol{x}_t)$ is conditioned on the true observation $\boldsymbol{x}_t$, observation reconstruction is an auto-encoding task. Only the KL-term corresponds to the future prediction task. On the other hand, in our case the *prior* latent state is decoded to a representation of the observation, which already constitutes a future prediction task. The KL-term only provides additional regularization. This makes the model less reliant on careful tuning of $\beta$ (empirically validated in Appendix C.3). In fact, setting it to zero also leads to reasonable performance in our experiments, though not the best.

## 2.3 Behavior Learning

Behavior is learned simultaneously with the state representation using SAC. This involves learning an actor $\pi_\phi$ that parameterizes a stochastic policy and two critics $Q_{\theta_i}, i \in \{1, 2\}$ which use the latent

state $\boldsymbol{z}_t$ as the representation of the agent's state. The critic is trained by minimizing Bellman error,

$$J_Q(\theta_i, \Theta) = \mathbb{E}_{\boldsymbol{z}_{t+1} \sim q} \left[ (Q_{\theta_i}(\boldsymbol{z}_t, \boldsymbol{a}_t) - \text{sg}(r_t + \gamma V_{\text{target}}(\boldsymbol{z}_{t+1})))^2 \right], \tag{3}$$

$$V_{\text{target}}(\boldsymbol{z}_{t+1}) = \mathbb{E}_{\boldsymbol{a}_{t+1} \sim \pi_\phi} \left[ \min_{i=1,2} Q_{\bar{\theta}_i}(\boldsymbol{z}_{t+1}, \boldsymbol{a}_{t+1}) - \alpha \log \pi_\phi(\boldsymbol{a}_{t+1}|\boldsymbol{z}_{t+1}) \right], \tag{4}$$

where $\bar{\theta}_i$ represents an exponentially moving average of $\theta_i$, 'sg' denotes the stop-gradient operation, and $\alpha$ is the weight on the entropy regularization. Note that $J_Q$ is a function of $\Theta$ as well. In other words, the RSSM and observation encoder model are also trained using the gradients from the critic loss. Appendix C.2 explores this design choice in more detail. The parameters $\Theta$ are shared between the two critics. The actor is updated using the following loss function

$$J_\pi(\phi) = \mathbb{E}_{\boldsymbol{z}_t \sim q, \boldsymbol{a}_t \sim \pi_\phi} [\alpha \log \pi_\phi(\boldsymbol{a}_t|\boldsymbol{z}_t) - \min_{i=1,2} Q_{\theta_i}(\boldsymbol{z}_t, \boldsymbol{a}_t)]. \tag{5}$$

Overall training consists of three steps: collecting data using the current actor, using the data to update the representation model, and using the data to update the actor and critic as shown in Algorithm 1.

## 3 EXPERIMENTS

We use the Distracting Control Suite (DCS) (Stone et al., 2021) and Robosuite (Zhu et al., 2020) to benchmark our model's ability to withstand visual distractions. DCS consists of six simulated robotic control tasks derived from the DeepMind Control Suite (Tassa et al., 2018). There are three types of distraction: background, camera, and color. They can either be fixed for the duration of the episode (static) or changed smoothly during the episode (dynamic). Three benchmarks are defined: easy, medium, and hard, which have increasing levels of difficulty along all 3 distraction types. We focus our experiments on two settings: dynamic-background-only and dynamic-medium. Results on other settings follow the same trends (Appendix B). In the dynamic-background-only case, the backgrounds for training and validation environments are drawn

---

**Algorithm 1:** The CoRe Model

**Require:** Environment E, initial $\Theta, \phi, \theta_1, \theta_2$
**for** *each iteration* **do**
    $\boldsymbol{o}_1 \sim E_{\text{reset}}(), \boldsymbol{z}_0 \leftarrow \boldsymbol{0}, \boldsymbol{a}_0 \leftarrow \boldsymbol{0}$;
    $\mathcal{D} \leftarrow \{\}$;
    **for** *each environment step* **do**
        $\boldsymbol{x}_t \leftarrow f_\theta(\boldsymbol{o}_t)$;
        $\boldsymbol{z}_t \leftarrow \text{RSSM}(\boldsymbol{z}_{t-1}, \boldsymbol{a}_{t-1}, \boldsymbol{x}_t)$;
        $\boldsymbol{a}_t \sim \pi_\psi(\boldsymbol{a}_t|\boldsymbol{z}_t)$;
        $r_t, \boldsymbol{o}_{t+1} \leftarrow E_{\text{step}}(\boldsymbol{a}_t)$;
        $\mathcal{D} \leftarrow \mathcal{D} \cup (\boldsymbol{o}_t, \boldsymbol{a}_t, r_t)$;
    **end**
    **for** *each gradient step* **do**
        $B \sim N$ samples of length $T$ from $\mathcal{D}$;
        Train the model: $\mathbb{E}_B[J_M(\Theta)]$;
        Train the critic: $\mathbb{E}_B[J_Q(\theta_i, \Theta)]$;
        Train the actor: $\mathbb{E}_B[J_\pi(\phi)]$;
    **end**
**end**

---

from the 60 training and 30 validation videos in the DAVIS dataset (Pont-Tuset et al., 2017). Our experiments show that (1) CoRe enables visual robotic control in the presence of severe distractions, performing better than strong baselines, (2) all three elements of CoRe - recurrence, contrastive learning, and dynamics modeling are important, (3) recurrent architectures are important to make contrastive learning work well, (4) CoRe can solve a challenging simulated robotic manipulation task, and (5) interpretable attention masks emerge that help visualize how CoRe works. We report the mean validation reward over 10 random seeds and 100 episodes per seed along with the standard error. Details of the network architectures and training hyper-parameters are provided in appendix A.

### 3.1 COMPARISON OF ROBUSTNESS TO DISTRACTIONS

This set of experiments quantifies the robustness of CoRe to visual distractions. We compare to a model-free RL baseline **SAC-RAD**, a state-of-the-art method based on bisimilarity metrics **PSE** (Agarwal et al., 2021), **Recon**, a reconstruction variant of our model, where we replace the contrastive loss with an observation reconstruction loss (making it similar to SLAC), and **CURL** (Laskin et al., 2020a) which also uses contrastive learning but does not use recurrence or future prediction. Figure 2 shows the comparison as a function of environment steps for 4 of the 6 environments. We use the published code for PSE and CURL and only change the distraction settings.

In the **no-distraction** setting (Figure 2, top row), we can see that all methods perform well. In the **dynamic-background-only setting** (Figure 2, middle row), CoRe and PSE both perform well overall

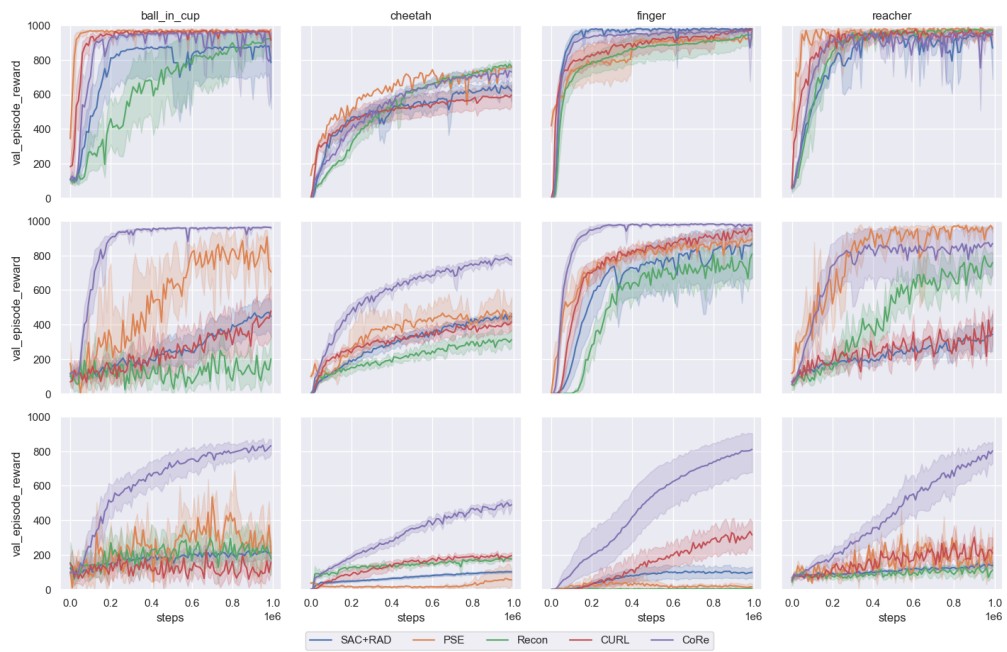

Figure 2: Results on pixel-based control tasks from the Distracting Control Suite in various distraction settings. **Top**: No distractions. **Middle**: Background distractions only. **Bottom** Background, color, and camera distractions.

| Method | Env Steps | Video dataset | BiC-Catch | C-swingup | C-run | F-spin | R-easy | W-walk |
|---|---|---|---|---|---|---|---|---|
| DBC | 500K | Kinetics (Driving) | – | $650 \pm 40$ | $260 \pm 40$ | $800 \pm 10$ | $100 \pm 20$ | $550 \pm 50$ |
| DBC | 500K | DAVIS (2 videos) | $170 \pm 57$ | $198 \pm 11$ | $94 \pm 6$ | $443 \pm 15$ | $74 \pm 11$ | $158 \pm 9$ |
| PSE | 500K | DAVIS (2 videos) | $\mathbf{821 \pm 17}$ | $\mathbf{749 \pm 19}$ | $308 \pm 12$ | $779 \pm 49$ | $\mathbf{955 \pm 10}$ | $\mathbf{789 \pm 28}$ |
| CURL | 500K | DAVIS (2 videos) | $580 \pm 53$ | $624 \pm 38$ | $313 \pm 20$ | $\mathbf{851 \pm 19}$ | $416 \pm 68$ | $773 \pm 35$ |
| CoRe (Ours) | 500K | DAVIS (2 videos) | $802 \pm 15$ | $578 \pm 32$ | $\mathbf{483 \pm 19}$ | $743 \pm 22$ | $610 \pm 96$ | $773 \pm 16$ |
| CoRe (Ours) | 1M | DAVIS (2 videos) | $784 \pm 25$ | $635 \pm 23$ | $527 \pm 26$ | $752 \pm 20$ | $620 \pm 90$ | $775 \pm 17$ |
| CURL | 500K | DAVIS (60 videos) | $220 \pm 42$ | $697 \pm 20$ | $332 \pm 17$ | $847 \pm 14$ | $302 \pm 29$ | $750 \pm 41$ |
| PSE | 500K | DAVIS (60 videos) | $667 \pm 21$ | $762 \pm 10$ | $392 \pm 6$ | $829 \pm 4$ | $\mathbf{943 \pm 6}$ | $718 \pm 81$ |
| CoRe (Ours) | 500K | DAVIS (60 videos) | $\mathbf{957 \pm 3}$ | $\mathbf{810 \pm 9}$ | $\mathbf{659 \pm 13}$ | $\mathbf{978 \pm 4}$ | $843 \pm 83$ | $\mathbf{880 \pm 23}$ |
| CoRe (Ours) | 1M | DAVIS (60 videos) | $961 \pm 4$ | $821 \pm 15$ | $773 \pm 14$ | $975 \pm 6$ | $869 \pm 81$ | $950 \pm 6$ |

Table 1: Background distractions only. Mean $\pm$ Standard Error. **Bold** indicates best results at 500K steps.

and better than the other methods. CoRe does significantly better than PSE on the 'cheetah-run' task. SAC-RAD does not perform well, confirming the finding in Stone et al. (2021) that data-augmented model-free RL methods are insufficient to solve these tasks. Recon also does not work well, as expected, because it wastes capacity on modeling the background. In the **dynamic-medium setting** (Figure 2, bottom row), we see that the performance of CoRe is far better than other baselines. CoRe is able to solve most of the tasks in this challenging distraction setting even while other methods struggle to make progress.

In Table 1, we compare the performance of our proposed model with bisimulation-based methods (DBC and PSE) and CURL. Results from DBC and PSE have been reported with different background videos making them hard to compare directly. DBC results were reported using videos from the Car Driving class in the Kinetics dataset (Kay et al., 2017), while PSE results were reported when training on 2 videos from DAVIS and testing on 30 validation videos. To do a fair comparison, we train CoRe as well as DBC (using their published code) in the same setting as PSE. This setting is denoted as DAVIS (2 videos) in Table 1. We can see that here, PSE outperforms DBC. CoRe performs similar to PSE. However, when we use all 60 training videos, CoRe performs better.

## 3.2 ABLATION EXPERIMENTS

In this set of experiments, we compare CoRe to the variants shown in Table 2 in the dynamic-medium distraction setting. From Table 3, we can see that model-free methods, SAC and QT-Opt, trained

| Attribute | SAC/QT-Opt | RSAC | RSAC+Dyn | Recon | CURL | RCURL | CoRe |
|---|---|---|---|---|---|---|---|
| Recurrence | - | ✓ | ✓ | ✓ | - | ✓ | ✓ |
| Contrastive Loss | - | - | - | - | ✓ | ✓ | ✓ |
| Dynamics Modeling | - | - | ✓ | ✓ | - | - | ✓ |

Table 2: Variants produced by ablating the three major elements of CoRe's design. RSAC and RCURL are variants of SAC and CURL which use a recurrent architecture (same as CoRe's). RSAC+Dyn models dynamics in the latent state space. Recon does the same, along with pixel reconstruction. All models use data augmentation.

| Method | Env Steps | Mean | BiC-Catch | C-swingup | C-run | F-spin | R-easy | W-walk |
|---|---|---|---|---|---|---|---|---|
| SAC + RAD[†] | 500K | $89 \pm 5$ | $139 \pm 7$ | $192 \pm 6$ | $14 \pm 2$ | $63 \pm 24$ | $93 \pm 6$ | $31 \pm 2$ |
| QT-Opt + RAD[†] | 500K | $103 \pm 3$ | $132 \pm 20$ | $241 \pm 7$ | $52 \pm 3$ | $25 \pm 6$ | $105 \pm 10$ | $64 \pm 2$ |
| RSAC | 500K | $198 \pm 20$ | $257 \pm 89$ | $277 \pm 19$ | $244 \pm 17$ | $90 \pm 47$ | $112 \pm 17$ | $211 \pm 15$ |
| RSAC + Dyn | 500K | $244 \pm 26$ | $175 \pm 74$ | $\mathbf{430 \pm 39}$ | $270 \pm 32$ | $66 \pm 54$ | $141 \pm 28$ | $379 \pm 55$ |
| Recon | 500K | $147 \pm 14$ | $211 \pm 55$ | $219 \pm 12$ | $147 \pm 9$ | $3 \pm 1$ | $105 \pm 5$ | $199 \pm 26$ |
| CURL | 500k | $223 \pm 14$ | $136 \pm 24$ | $320 \pm 11$ | $170 \pm 10$ | $163 \pm 37$ | $222 \pm 37$ | $328 \pm 19$ |
| RCURL | 500k | $232 \pm 20$ | $318 \pm 79$ | $280 \pm 11$ | $216 \pm 9$ | $154 \pm 65$ | $106 \pm 5$ | $321 \pm 13$ |
| CoRe (Ours) | 500K | $\mathbf{480 \pm 23}$ | $\mathbf{706 \pm 39}$ | $354 \pm 26$ | $\mathbf{354 \pm 10}$ | $\mathbf{540 \pm 73}$ | $\mathbf{445 \pm 48}$ | $\mathbf{479 \pm 31}$ |
| CoRe (Ours) | 1M | $684 \pm 24$ | $832 \pm 22$ | $483 \pm 29$ | $490 \pm 13$ | $810 \pm 60$ | $801 \pm 32$ | $686 \pm 42$ |

Table 3: Results on Distracting Control Suite Benchmark (dynamic-medium setting). Mean ± Standard Error. **Bold** indicates best results at 500K steps. † indicates baselines reported in Stone et al. (2021).

with data augmentation are not able to solve these tasks. Adding recurrence (RSAC) and dynamics modeling (RSAC+Dyn) improves results. However, trying to predict pixels hurts performance (Recon). Contrastive prediction in a learned space (CoRe) works much better. CURL and RCURL, which use a constrastive loss but do not model dynamics, perform worse that CoRe, showing that just having a contrastive loss is insufficient. More ablation experiments that study the relative effects of modeling rewards, forward and inverse dynamics are presented in appendix C.1.

### 3.3 IMPORTANCE OF HAVING A RECURRENT STATE SPACE MODEL

Contrastive models have been reported to underperform DBC (Zhang et al., 2021). However, in our experiments we found that contrastive models work quite well. We hypothesize that this difference is because we use a recurrent state space whereas previous work used contrastive learning with a feed-forward encoder operating on stacked observation frames. To validate this hypothesis, we compare our model to its non-recurrent version. The models both do contrastive future prediction and use similar network architectures, but the non-recurrent one consumes three stacked frames. The training mini-batches are sampled similarly using 32 sequences of 32 time-steps each. Therefore, even the non-recurrent model has access to observations that are semantically close. The results at 500K environment steps in the dynamic-medium setting on DCS are shown in Figure 3 (Top). We can see that, in the absence of recurrence, contrastive learning does not perform well. This indicates that just having nearby observations is not sufficient to provide hard negatives and a recurrent network has help fix this problem.

An intuitive explanation for this is the following. A recurrent network (in our case, a GRU-RNN) has a smoothness bias built-in because at each time step, it carries forward previous state and only modifies it slightly by gating and adding new information. This is true to a large extent even during training, and not just at convergence. Therefore, when CoRe is trained, it generates hard negatives through-

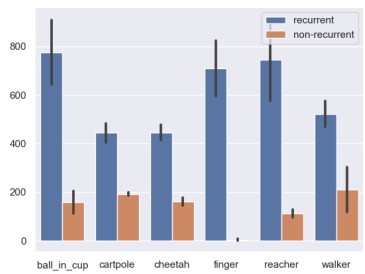

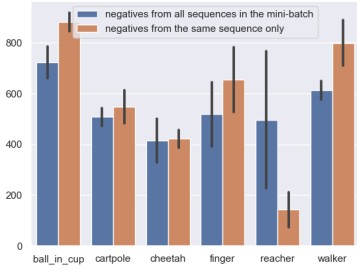

Figure 3: **Top:** Scores drop if recurrence is removed. **Bottom:** Using negatives only from the same episode is often helpful.

out training in the form of nearby future predictions. This is true even when the observations have distractions present which change the overall appearance of the observation dramatically. On the other

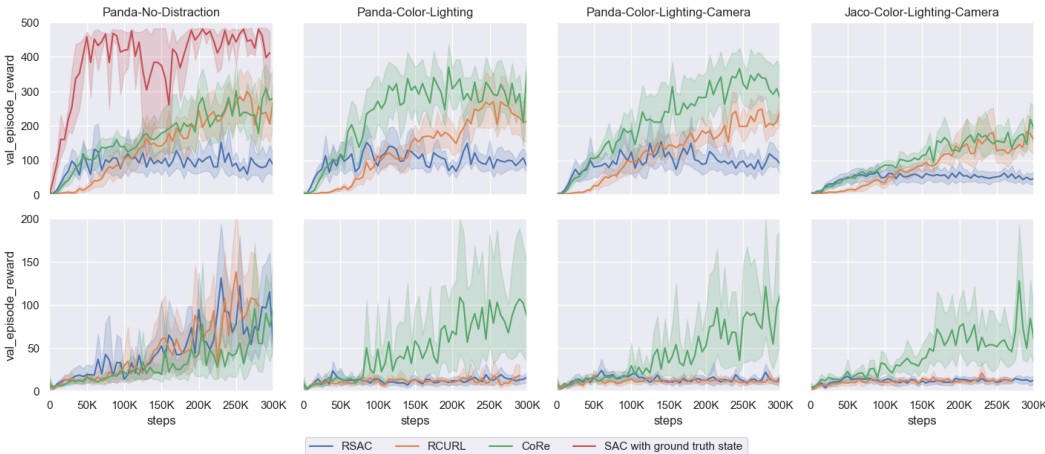

Figure 4: Results on the Door Opening Task Robosuite (Zhu et al., 2020). Each column corresponds to a different distraction setting or robot arm type. **Top**: Camera observation as well as the proprioceptive state is provided to the agent. **Bottom**: Only camera observation is provided, requiring the vision system to do more work.

hand, starting from a random initialization, feed-forward networks are less likely to map observations that are semantically close but visually distinct to nearby points. Therefore, hard negatives may not be found easily during training.

To confirm this further, we train CoRe with a modified contrastive loss in which for each sample in the mini-batch, only the same temporal sequence is used to sample negatives. As shown in Figure 3 (bottom), this is not harmful (but actually beneficial) to performance on all tasks, except reacher. This means that for most tasks, CoRe doesn't need other sequences in the mini-batch to find hard negatives. This avoids a major concern in contrastive learning, which is having large mini-batches to ensure hard negatives can be found. Essentially, recurrence provides an architectural bias for generating good negative samples locally. Performance degrades on the reacher task because observations there contain a goal position for the arm. Contrastive learning tries to remove this information because it is constant throughout the episode. Therefore, the actor and critic may not get access to the goal information, causing the agent to fail. This highlights a **key limitation** of contrastive learning – it discourages retaining constant information in the latent space. For that type of information, it is important to have negative examples coming from other episodes.

## 3.4 RESULTS ON ROBOSUITE

In this section, we present results of our method on Robosuite (Zhu et al., 2020), a MuJoCo-based simulator for robotic manipulation tasks. We experiment with two kinds of arms: Panda and Jaco, and two distraction settings: (1) Color and lighting only, and (2) Color, lighting, and camera. In this experiment, we use the static distraction setting to emulate a scenario where data collected within an episode has consistent lighting, color, and camera position, but these factors vary across episodes (depending on time of day, accidental movement of the camera relative to the robot setup, etc). The action space in delta end-effector pose. Control is run at 20Hz, with each episode lasting for 500 steps (25 seconds). Architecture and training details are described in appendix A.

Figure 4 compares CoRe with recurrent versions of SAC and CURL. In the distraction-free case, we also compare to results from SAC operating on ground-truth object state instead of using image observations, as reported by Zhu et al. (2020). This can be seen as an upper bound on performance because the agent gets perfect access to task-relevant state. In the case where proprioception (the robot's state) is provided separately (top row), the performance of CoRe is same as or better than that of CURL. In the harder case where the robot's state must also be obtained from the camera observation, CoRe significantly outperforms other methods. This shows that CoRe is better at handling more complex vision tasks. Comparison to other baselines is included in appendix C.4.

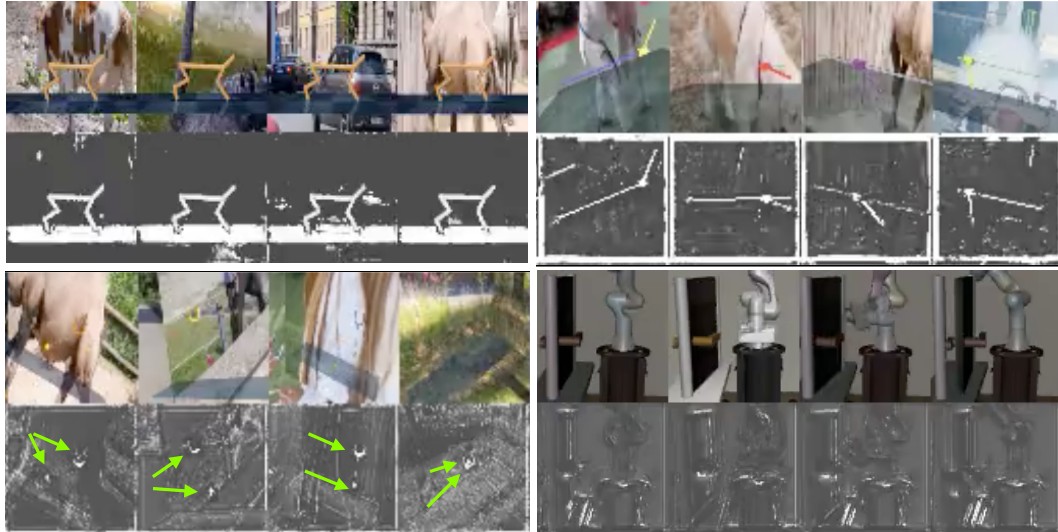

Figure 5: Gating masks showing that CoRe learns to remove distractions. Green arrows indicate the locations of the ball and cup, which are tiny, but CoRe can still find them. In the door opening task in Robosuite (bottom right), the lower edge of the door handle is very bright when the door is closed, indicating strong attention to that part of the image. In addition, the agent attends to the edges of the door and the table, as well as its own base. Since the camera position can change randomly, the agent needs to find the robot's position relative to the door, for which the robot's base and the door's edges are important. Proprioception is provided separately. Therefore, the agent doesn't need to attend to the arm itself.

## 3.5 Visualizations

In this section, we present qualitative visualizations to help understand what the model does to cope with distractions. We make visualization easier by modifying the observation encoder to include a pixel-level gating network. This is done using a four-layered CNN which takes the observation as input and outputs a full-resolution sigmoid mask. This mask is multiplied into the observation image, and the result is fed to the encoder. The gating network is trained along with the rest of the model. We expected this gating to improve performance but found that it actually has no effect. This lets us use it as a diagnostic tool that factors out the attention mask that the model is otherwise applying implicitly. Figure 5 shows the attention masks inferred by the gating network. We can see that the model learns to exclude distractions in the background, even in the dynamic-medium distraction setting for the ball_in_cup task where the task-relevant objects are very small. For the door opening task, the model learns to attend to the bottom edge of the door handle, the top and right edges of the door, the edge of the table, and some parts of the robot's base. This includes all the information needed to find the relative position of the door with respect to the agent. Proprioception is provided separately. Therefore, the agent doesn't need to attend to the arm itself.

## 4 Discussion and Related work

Our work uses contrastive learning in the context of world modeling and pixel-based robotic control. In this section, we review related work in these areas.

**Pixel-based robotic control** Controlling robots directly from raw environment observations is a general framework for robotic problem solving since it avoids the use of specific modules such as object detectors and instead relies on the data and reward function to learn a representation of the observations. Model-free methods such as QT-Opt (Kalashnikov et al., 2018) and SAC (Haarnoja et al., 2018) have been applied to pixel-based control. However, learning encoders for high-dimensional observations from reward only is sample inefficient. Simply adding observation autoencoding to SAC as an auxiliary task has been shown to work well in Yarats et al. (2019). Instead of using a generative model, CURL (Laskin et al., 2020a) uses contrastive learning to learn observation encoders yielding similar results. ATC (Stooke et al., 2021) improves on CURL by predicting representations for future observations, instead of the current one. Data augmentation techniques RAD (Laskin et al., 2020b) and DrQ (Yarats et al., 2021) dramatically improve sample efficiency. However, these methods are

still not sufficient to solve visual tasks in the presence of distractions (Stone et al., 2021). Other approaches shape the learned encoders to produce a latent space that is amenable to locally-linear control algorithms (Watter et al., 2015; Cui et al., 2020; Levine et al., 2020). Improvements in the encoder architecture, for example, using attention Tang et al. (2020) and efficient transformer-based models (Choromanski et al., 2021) have been proposed. Our proposed model can potentially benefit from the use of these better architectures. We use a simple convolutional encoder for our experiments and focus on the modeling aspect of the problem.

**World Modeling** World models aim to learn how the world changes as a result of the agent's action instead of just learning the optimal policy. Finn et al. (2016) used video prediction parameterized as pixel-level flow to predict the future video frames. Dreamer (Hafner et al., 2020) and SLAC (Lee et al., 2020) model observations using a sequential action-conditioned VAE. However, these methods are not designed to deal with distractions since they would waste capacity on reconstructing them along with the task-relevant portion of the observations. PBL (Guo et al., 2020) improves on Dreamer using an additional term that predicts the state given observations. In order to avoid observation reconstruction, Gelada et al. (2019) proposed DeepMDP, a method that embeds the given high-dimensional RL problem into a latent space which preserves the essence of the problem (i.e. the reward and transition function). While this method only preserves the current reward, Deep Bisimulation for Control (Zhang et al., 2021) uses the future sequence of rewards to extract a stronger learning signal. In this case, the learned state space is trained to impose a bisimulation metric, under which similarity between states is determined by the similarity in future reward sequences produced from those states. PSE (Agarwal et al., 2021) further extends this idea to use similarity in future action distributions. While these models use dynamics to get more training signal, they do not make use of structure in the observation space itself. In our proposed model, we seek to find a middle ground between modeling observations and not modeling them at all. This is done in a principled way using mutual information maximization between the observation and its prediction from the internal dynamics. While previous work often found contrastive learning to be ineffective, we show that combining it with recurrent state space models makes it work. Recently, a contrastive variant of Dreamer (Okada & Taniguchi, 2020) has been proposed which shares the same motivation. Concurrent with our work, Nguyen et al. (2021) explore a formulation similar to ours based on temporal predictive coding, but do not evaluate it on the difficult camera and color distractions we do here.

**Contrastive Learning**: Learning to contrast positive and negative examples is a general representation learning tool. It can be applied in many domains, for example, in face verification (Weinberger et al., 2006; Schroff et al., 2015), object tracking (Wang & Gupta, 2015), time-contrastive models (Sermanet et al.; Hyvarinen & Morioka, 2016), language models such as Log-Bilinear models Mnih & Hinton (2007), Word2Vec (Mikolov et al., 2013) and Skip-thoughts (Kiros et al., 2015). Contrastive Predictive Coding (CPC) (van den Oord et al., 2018) proposes an auto-regressive model to do future prediction at multiple time steps and applies it to various domains including image patches, audio, and RL. CPC|A (Guo et al., 2019) adds action-conditioning to the CPC model. Our model uses a similar contrastive loss as CPC|A. However, it also uses latent state correction using the KL term (Equation 2). Subsequent works have improved on CPC using momentum in the learned encoder (MoCo (He et al., 2020)), adding projection heads before computing contrastive loss (SimCLR (Chen et al., 2020)), avoiding negative examples altogether (BYOL (Grill et al., 2020) and SwAV (Caron et al., 2020)). Our model can potentially be improved using these techniques but we found that the simple contrastive loss is already sufficient to solve challenging visual control problems. Most similar to our work is CVRL (Ma et al., 2020) which uses contrastive learning with a recurrent architecture but with the standard sequential VAE formulation instead of the one with prior decoding that is used in CoRe.

## 5 Conclusion

We show that a contrastive objective can be a viable replacement for reconstructing observation pixels in solving visual robotics control problems using model-based RL. It works even in the presence of severe distractions. In addition, we show that having a recurrent architecture helps contrastive learning work well. Even though our results are only in simulated environments, the surprisingly high degree of robustness indicates that this approach should transfer well to a real-world setting. In future work, we plan to test this approach on real-world robots.

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

## A  ARCHITECTURE AND TRAINING DETAILS

In this section, we present the architecture and training hyper-parameters for the proposed CoRe model in more detail.

### A.1  GRU-RNN WITH STOCHASTIC AND DETERMINISTIC STATES

CoRe uses a recurrent state-space model (RSSM) that is based on GRUs (Cho et al., 2014) and similar to the one used in Dreamer (Hafner et al., 2020), as shown in the figure below. Latent states at any time-step consist of a (deterministic) recurrent hidden state and a stochastic state. The prior latent state is $\hat{z}_t = [h_t, \hat{s}_t]$ and the posterior latent state is $z_t = [h_t, s_t]$. They share the same recurrent hidden state $h_t$ but differ in the stochastic component. $\hat{s}_t$ depends on $h_t$, whereas $s_t$ depends on $h_t$ and the new observation features $x_t$. The feed-forward operation is

$$
\begin{aligned}
h_t &= \text{GRUCell}(h_{t-1}, \text{GRU\_MLP}(s_{t-1}, a_{t-1})), \\
\hat{\mu}, \hat{\sigma} &= \text{PriorMLP}(h_t), \\
\hat{s}_t &= \hat{\mu} + \hat{\sigma} * \mathcal{N}(0, 1), \\
\mu, \sigma &= \text{PosteriorMLP}(h_t, \text{ObsMLP}(x_t)), \\
s_t &= \mu + \sigma * \mathcal{N}(0, 1),
\end{aligned}
$$

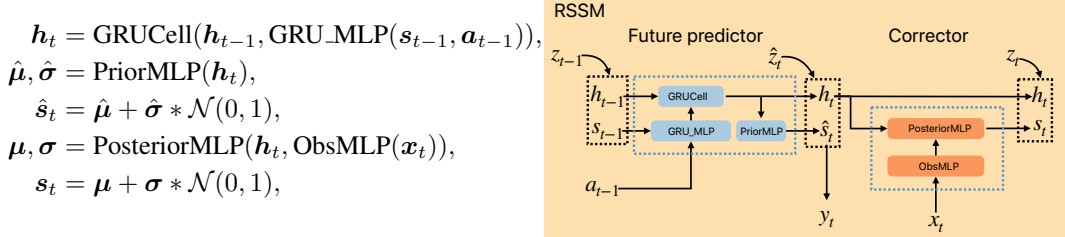

where MLP stands for a fully-connected multi-layered neural network. The figure above illustrates this operation. It can be seen as a more detailed view of the RSSM shown in Figure 1(b). Note that the recurrent hidden state $h_t$ is propagated through time without sampling. This is important because sampling can make it hard to retain information for long time-scales and destroys smoothness of the latent state. However, information inserted into the RNN through $s_t$ is stochastic, making the RSSM capable of generating multiple futures. This prevents mode-averaging and allows the model to generate diverse trajectories when doing rollouts.

### A.2  ARCHITECTURE

The model consists of a number of components: observation encoder CNN, RSSM (which includes PriorMLP, PosteriorMLP, GRU\_MLP, GRUCell, ObsMLP, along with the reward, inverse dynamics, and observation representation decoder networks), and the actor and critic networks. Table 4 describes the architecture of these networks. CNN layers are denotes as [num\_filters, kernel\_size, stride]. We borrow the observation encoder architecture from previous work (Laskin et al., 2020a; Yarats et al., 2019) and use it as-is to ensure an accurate comparison. ELU non-linearity (Clevert et al., 2016) is used in all places except the observation encoder (which uses ReLU units).

For experiments with Robosuite, the observation encoder architecture is modified to accept $128 \times 128$ resolution images. Two convolutional layers with 32 filters of kernel size $3 \times 3$ with a stride of 2 were applied to reduce the spatial size from $128 \times 128$ to $32 \times 32$. This was followed by 3 layers of $3 \times 3$ convolutions with a stride of 1.

### A.3  TRAINING

The training is broken into iterations, where in each iteration one episode of data are collected, added to the replay buffer, and a number of gradient steps are taken using mini-batches sampled from the replay buffer.

**Data Collection** For the first 1000 steps, data are collected by taking actions sampled from a uniform distribution in $[-1, 1]$. After that, data are collected by taking actions sampled from the learned actor policy, which requires rolling out the RSSM. At each training iteration, an entire episode of data are collected, where the length of the episode is 1000 steps of the underlying MuJoCo simulator. The actual number of steps is 1000 divided by the number of times the same action is repeated, which is standard based on the task (Table 6). We rollout an entire episode because we use a recurrent model and it seemed natural to use the same model through time within an episode. Typically, model-free RL algorithms collect data one environment step at a time. We did not explore per-step data collection

| Component | Architecture |
|---|---|
| Observation Encoder | CNN : $[32, 3\times3, 2]$, $[32, 3\times 3, 1]\times3$, 50 fully-connected, layer norm |
| PriorMLP | $[400, 400, 400]$ |
| PosteriorMLP | $[400, 400]$ |
| GRUCell | 200 hidden units |
| GRU_MLP | $[400, 400]$ |
| ObsMLP | $[256, 256]$, layer norm |
| Reward prediction | $[128, 128, 1]$ |
| Inverse dynamics prediction | $[128, 128, a_{\text{dims}}]$ |
| Observation representation decoder | $[256, 256, 50]$, layer-norm |
| Actor | $[1024, 1024, 1024, 2a_{\text{dims}}]$ |
| Critic | $[1024, 1024, 1024, 1]$ |
| Gating (optional) | CNN :$[16, 5 \times 5, 1] \times 3$, $[1, 1 \times 1, 1]$, sigmoid. |

Table 4: Architecture of model components.

| Parameter | Value |
|---|---|
| Replay Buffer | 100,000 |
| Initial steps | 1000 |
| Model Learning rate | 3.e-4 |
| Critic Learning rate | 1.e-3 |
| Actor Learning rate | 1.e-3 |
| Target entropy | $-a_{\text{dims}}$ |
| Actor update frequency | 1 |
| Target critic update frequency | 1 |
| Target critic update $\tau$ | 0.005 |
| Action log std range | $[-10, 2]$ |
| KL-weight $\beta$ | 0.01 |
| Reward prediction weight $\alpha_r$ | 1.0 |
| Inverse Dynamics weight $\alpha_a$ | 1.0 |
| Weight decay | 0 |
| Critic max grad norm clip | 100 |
| Actor max grad norm clip | 10 |
| Model max grad norm clip | 10 |

Table 5: List of hyper-parameters.

in this work, although that could work just as well. The data from the Distracting Control Suite is rendered at a $320 \times 240$ resolution which is resized to $100 \times 100$. Pixels are normalized to $[0, 1]$ by dividing by 255. All actions are normalized to lie in $[-1, 1]$ and are modeled using tanh-squashed Gaussian distributions. For robosuite, visual observations were rendered from the front-view camera at a $256 \times 256$ resolution and a $128 \times 128$ crop near the center of the image was fed as input to the model.

**Model updates** Training is done with mini-batches of $N = 32$ sequences of length $T = 32$ each sampled from the replay buffer. Data augmentation is done by taking random $84 \times 84$ crops. The same crop position is used across the entire sequence. Each mini-batch is used to do three updates sequentially corresponding to the three losses: $J_M$, $J_Q$ and $J_\pi$. The number of updates is chosen to be 0.5 times the number of steps in an episode. All training hyper-parameters are listed in Table 5.

**Implementation** The model is implemented using PyTorch (Paszke et al., 2019). The environment is based on MuJoCo (Todorov et al., 2012). All training was done on single Nvidia A100 GPUs. Training time for 500K updates varies from 12-24 hr depending on the task. Training times differ due to different values of action repeat. Our implementation will be made public.

## A.4 TRAINING CURVES FOR INDIVIDUAL LOSS TERMS

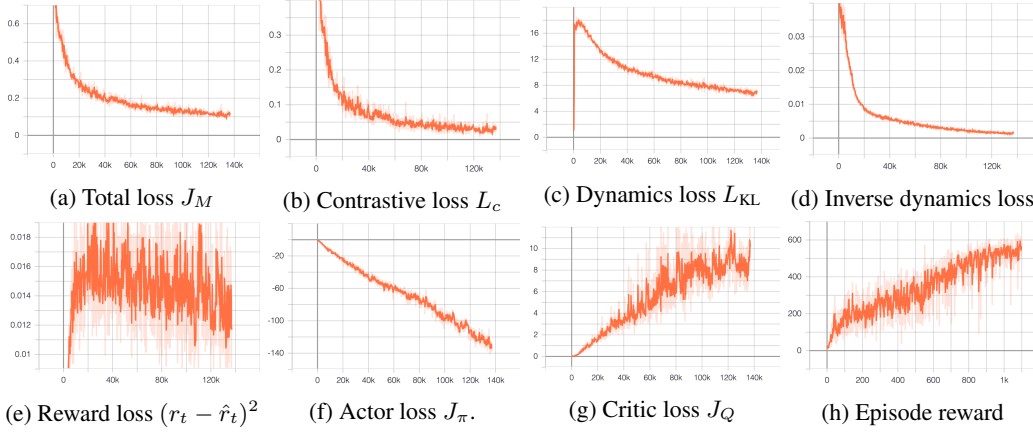

(a) Total loss $J_M$  (b) Contrastive loss $L_c$  (c) Dynamics loss $L_{\text{KL}}$  (d) Inverse dynamics loss

(e) Reward loss $(r_t - \hat{r}_t)^2$  (f) Actor loss $J_\pi$.  (g) Critic loss $J_Q$  (h) Episode reward

Figure 6: Training curves for a typical run of CoRe training on cheetah, dynamic-medium setting.

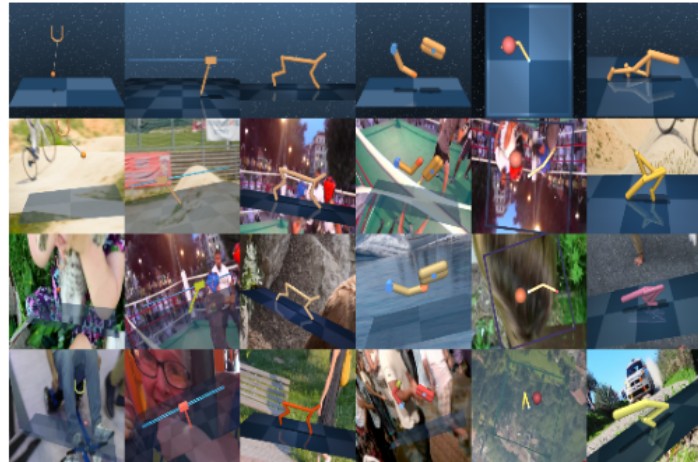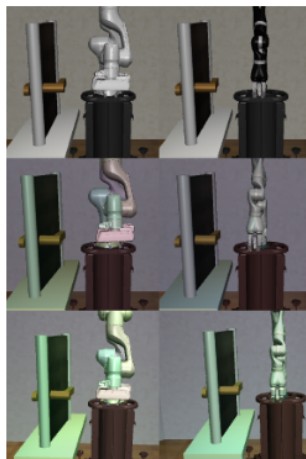

Figure 7: **Left**: Examples from the Distracting Control Suite (Stone et al., 2021). The top row shows the clean versions of each of the six tasks. The subsequent rows show examples from the easy, medium, and hard distraction settings. **Right**: Examples from the Robosuite door opening task (Zhu et al., 2020). The top row shows the clean versions of the tasks with a Panda and Jaco arm. The middle row shows observations with color and lighting distractions. The bottom row adds on camera distraction.

There are a number of loss terms which are linearly combined to create the total model loss $J_M$. In this section, we present training curves that show how these individual terms optimize with training. Figure 6 shows these loss terms, along with the RL training losses ($J_Q$ and $J_\pi$) and the episode reward. We can see that the total loss $J_M$ goes down as expected, along with the individual terms. Note that the reward loss is low in the beginning because the agent gets zero rewards which is easy for the model to predict. As the agent starts receiving better rewards, the reward error increases but then eventually starts to come down. The KL-term has a similar behavior. At the very beginning, the prior and posterior latent state distributions match each other (making their KL divergence small) but both are bad at modeling the data. As more training data is encountered, the KL rises sharply, then eventually reduces with more training. The actor loss (which is dominated by the negative of the Q-value of the actor's chosen action) goes down as expected, indicating that the actor outputs actions that have high Q-values. The critic loss appears to increase because the magnitude of the Q-values increases with training, causing the magnitude of the Bellman residual to go up as well.

## B   COMPLETE RESULTS ON DISTRACTING CONTROL SUITE

In this section, we report the performance of CoRe on all the distraction settings in the Distracting Control Suite. There are three difficulty levels: easy, medium, and hard[1] as shown in Figure 7. The difficulty is set by increasing the scale of the camera pose and color change and the number of background distractions that are used (Table 7). For each difficulty level, there are two settings: **static**, in which the distraction (a background image, or a particular choice of random camera pose) is fixed throughout the episode, and **dynamic** in which the distraction changes smoothly. In the case of background distractions, the video plays back-and-forth, ensuring no sharp cuts. In the case of the camera distractions, the camera moves along a smooth trajectory.

Table 8 compares the performance of CoRe with model-free RL baselines such as SAC (Haarnoja et al., 2018) and QT-Opt (Kalashnikov et al., 2018) combined with data augmentation techniques RAD (Laskin et al., 2020b) and DrQ (Yarats et al., 2021) as reported in (Stone et al., 2021). We also compare with a recurrent SAC+RAD baseline (RSAC) which uses the same recurrent architecture as CoRe, but does not contrastively predict the next observation, or model the dynamics and reward. Comparisons are made at 500K environment steps, though we report our results at 1M environment

---

[1]The 'hard' level is described in the code but results are not reported in the main paper (Stone et al., 2021).

Table 6: Action repeat

| Task | Action repeat |
|---|---|
| Ball in cup catch | 4 |
| Cartpole swingup | 8 |
| Cheetah Run | 4 |
| Finger spin | 2 |
| Reacher easy | 4 |
| Walker walk | 2 |

Table 7: Distraction settings in the Distracting Control Suite.

| Difficulty | Train videos | Val videos | Camera and Color change scale |
|---|---|---|---|
| Easy | 4 | 4 | 0.1 |
| Medium | 8 | 8 | 0.2 |
| Hard | 60 | 30 | 0.3 |

steps as well to show that our model continues to improve with more steps. Performance is averaged over 10 random seeds, and 100 validation episodes at each checkpoint. When no distractions are present (Table 8a), all methods perform well, though CURL and CoRe are slightly better than the rest in terms of mean scores. On the easy benchmark (Table 8b), CoRe outperforms other methods in all tasks, except reacher. As discussed in Section 3.3, this points to a key limitation of contrastive learning-based methods, which is that they tend to remove constant information (such as the goal location for reacher). However, at 1M steps, the performance on reacher is much better, showing that the model is able to eventually solve the task.

On the medium benchmark (Table 8c) CoRe outperforms other methods across all tasks, showing that it can deal with the presence of more distractions. The performance on the reacher tasks improves slightly over the easy setting, which shows that having more variation in the distractions actually helps training, whereas it hurts the baseline methods. We also report results on the hard setting (Table 8d), which is documented in the DCS codebase. Stone et al. (2021) do not report model-free baselines for this setting, presumably because the baseline models fail to train reasonable policies at all. However, CoRe is able to get off the ground and get reasonable performance even in this setting. Training to 1M steps continues to improve results.

Figure 8 shows the progression of validation reward for CoRe over 1M environment steps for all tasks and distractions settings. We can see that the hard-dynamic setting (green curves in the bottom two rows) is the hardest to fit because the performance increases slowly. However, in most cases the performance for that setting is still improving at 1M steps, and is likely to get better with more training. Our model struggles on the reacher task in terms of performance reported at 500K and even 1M steps, but we can see from these plots that the model is likely to continue improving if trained beyond 1M steps in both static and dynamic settings. In our experiments, we did not tune hyper-parameters specifically for each task, so it is possible that some tasks can benefit from further tuning. In particular, for the reacher task, a bigger batch-size can help since that is the only way to get access to diverse target positions.

## C    ADDITIONAL ABLATIONS

### C.1    IMPORTANCE OF INVERSE DYNAMICS AND REWARD PREDICTION

In addition to forward dynamics prediction, the proposed CoRe model includes reward prediction and inverse dynamics prediction as auxiliary tasks. In this section, we present comparisons to ablated versions of our model that remove one or both of these tasks. In Figure 9a we can see that removing both tasks (Fwd only) is significantly worse than having both (Fwd + inv + reward). Having any one of these alone is a big improvement on all tasks except reacher, where adding reward is much more important than inverse dynamics. It is interesting to see that in the absence of reward, adding inverse dynamics prediction (Fwd + inv) improves over having forward dynamics only. Asking the model to predict the action that takes the agent from one state to the next is a different way of expressing the model dynamics, compared to predicting the next state given the current state and action. The fact that asking the model to do both simultaneously gives a boost in performance indicates that inverse dynamics prediction shapes the latent state in ways that are complementary to forward dynamics.

### C.2    IMPORTANCE OF UPDATING Θ USING CRITIC LOSS

In our model we optimize the parameters $\Theta$ of the world model (observation encoder and RSSM) using the critic loss $J_Q$. This is similar to the choice made in SLAC (Lee et al., 2020) and DBC (Zhang

Table 8: Results on the Distraction Control Suite benchmark. All results reported at 500K steps, unless mentioned otherwise. Mean reward ± Standard Error. **Bold** numbers indicate the best performing models at 500K steps. † indicates baselines reported in Stone et al. (2021).

(a) No distraction

| Method | Mean | BiC-Catch | C-swingup | C-run | F-spin | R-easy | W-walk |
|---|---|---|---|---|---|---|---|
| SAC+RAD† | $836 \pm 29$ | $962 \pm 2$ | $843 \pm 10$ | $515 \pm 13$ | $\mathbf{976 \pm 5}$ | $962 \pm 8$ | $762 \pm 184$ |
| QT-Opt+RAD† | $820 \pm 3$ | $\mathbf{968 \pm 1}$ | $843 \pm 14$ | $538 \pm 11$ | $953 \pm 1$ | $969 \pm 5$ | $648 \pm 25$ |
| QT-Opt+DrQ† | $801 \pm 5$ | $962 \pm 2$ | $\mathbf{851 \pm 5}$ | $534 \pm 12$ | $952 \pm 1$ | $\mathbf{974 \pm 1}$ | $532 \pm 29$ |
| RSAC | $815 \pm 32$ | $873 \pm 80$ | $629 \pm 108$ | $527 \pm 28$ | $\mathbf{977 \pm 4}$ | $952 \pm 8$ | $934 \pm 5$ |
| CURL | $\mathbf{859 \pm 21}$ | $963 \pm 4$ | $\mathbf{850 \pm 3}$ | $534 \pm 37$ | $922 \pm 29$ | $949 \pm 13$ | $\mathbf{937 \pm 4}$ |
| CoRe | $\mathbf{853 \pm 19}$ | $953 \pm 2$ | $796 \pm 19$ | $\mathbf{628 \pm 16}$ | $950 \pm 17$ | $846 \pm 62$ | $\mathbf{942 \pm 7}$ |
| CoRe (1M steps) | $879 \pm 20$ | $811 \pm 97$ | $852 \pm 7$ | $730 \pm 15$ | $966 \pm 13$ | $963 \pm 7$ | $954 \pm 2$ |

(b) Benchmark easy

| | Method | Mean | BiC-Catch | C-swingup | C-run | F-spin | R-easy | W-walk |
|---|---|---|---|---|---|---|---|---|
| Static | SAC+RAD† | $182 \pm 24$ | $129 \pm 20$ | $360 \pm 25$ | $72 \pm 44$ | $370 \pm 114$ | $102 \pm 14$ | $60 \pm 31$ |
| | QT-Opt+RAD† | $317 \pm 8$ | $218 \pm 44$ | $446 \pm 23$ | $220 \pm 5$ | $711 \pm 27$ | $181 \pm 17$ | $128 \pm 14$ |
| | QT-Opt+DrQ† | $299 \pm 6$ | $217 \pm 35$ | $416 \pm 20$ | $199 \pm 8$ | $695 \pm 33$ | $171 \pm 25$ | $93 \pm 9$ |
| | RSAC | $274 \pm 25$ | $205 \pm 44$ | $491 \pm 36$ | $283 \pm 26$ | $92 \pm 53$ | $113 \pm 10$ | $459 \pm 18$ |
| | CURL | $418 \pm 32$ | $165 \pm 35$ | $430 \pm 30$ | $357 \pm 13$ | $759 \pm 19$ | $142 \pm 25$ | $657 \pm 47$ |
| | CoRe | $\mathbf{634 \pm 29}$ | $\mathbf{854 \pm 12}$ | $\mathbf{562 \pm 15}$ | $\mathbf{459 \pm 14}$ | $\mathbf{870 \pm 39}$ | $\mathbf{319 \pm 46}$ | $\mathbf{742 \pm 30}$ |
| | CoRe (1M steps) | $769 \pm 18$ | $876 \pm 15$ | $681 \pm 15$ | $596 \pm 32$ | $920 \pm 32$ | $666 \pm 34$ | $875 \pm 9$ |
| Dynamic | SAC+RAD† | $270 \pm 31$ | $366 \pm 59$ | $297 \pm 21$ | $198 \pm 39$ | $338 \pm 59$ | $173 \pm 11$ | $249 \pm 138$ |
| | QT-Opt+RAD† | $343 \pm 24$ | $490 \pm 64$ | $467 \pm 12$ | $170 \pm 8$ | $393 \pm 91$ | $\mathbf{428 \pm 68}$ | $109 \pm 12$ |
| | QT-Opt+DrQ† | $265 \pm 5$ | $395 \pm 39$ | $431 \pm 18$ | $126 \pm 10$ | $203 \pm 33$ | $343 \pm 53$ | $91 \pm 3$ |
| | RSAC | $275 \pm 24$ | $181 \pm 32$ | $465 \pm 21$ | $292 \pm 10$ | $86 \pm 55$ | $145 \pm 31$ | $482 \pm 20$ |
| | CURL | $391 \pm 30$ | $102 \pm 20$ | $432 \pm 15$ | $233 \pm 13$ | $648 \pm 32$ | $253 \pm 40$ | $678 \pm 35$ |
| | CoRe | $\mathbf{586 \pm 30}$ | $\mathbf{798 \pm 30}$ | $\mathbf{499 \pm 22}$ | $\mathbf{423 \pm 22}$ | $713 \pm 81$ | $340 \pm 60$ | $\mathbf{744 \pm 40}$ |
| | CoRe (1M steps) | $722 \pm 28$ | $909 \pm 10$ | $590 \pm 17$ | $569 \pm 19$ | $823 \pm 75$ | $552 \pm 83$ | $889 \pm 26$ |

(c) Benchmark medium

| | Method | Mean | BiC-Catch | C-swingup | C-run | F-spin | R-easy | W-walk |
|---|---|---|---|---|---|---|---|---|
| Static | SAC+RAD† | $113 \pm 12$ | $96 \pm 14$ | $272 \pm 11$ | $21 \pm 15$ | $169 \pm 92$ | $93 \pm 6$ | $25 \pm 1$ |
| | QT-Opt+RAD† | $165 \pm 15$ | $172 \pm 12$ | $297 \pm 7$ | $130 \pm 7$ | $234 \pm 67$ | $94 \pm 16$ | $63 \pm 3$ |
| | QT-Opt+DrQ† | $170 \pm 11$ | $169 \pm 25$ | $283 \pm 5$ | $124 \pm 9$ | $266 \pm 51$ | $112 \pm 16$ | $64 \pm 4$ |
| | RSAC | $199 \pm 20$ | $164 \pm 23$ | $428 \pm 17$ | $164 \pm 11$ | $9 \pm 5$ | $80 \pm 3$ | $350 \pm 15$ |
| | CURL | $300 \pm 28$ | $124 \pm 33$ | $304 \pm 20$ | $277 \pm 12$ | $621 \pm 21$ | $74 \pm 22$ | $402 \pm 78$ |
| | CoRe | $\mathbf{561 \pm 29}$ | $\mathbf{762 \pm 27}$ | $\mathbf{509 \pm 14}$ | $\mathbf{402 \pm 15}$ | $\mathbf{880 \pm 9}$ | $\mathbf{219 \pm 25}$ | $\mathbf{593 \pm 26}$ |
| | CoRe (1M steps) | $690 \pm 26$ | $743 \pm 88$ | $634 \pm 11$ | $526 \pm 17$ | $924 \pm 5$ | $543 \pm 56$ | $766 \pm 30$ |
| Dynamic | SAC+RAD† | $89 \pm 5$ | $139 \pm 7$ | $192 \pm 6$ | $14 \pm 2$ | $63 \pm 24$ | $93 \pm 6$ | $31 \pm 2$ |
| | QT-Opt+RAD† | $103 \pm 3$ | $132 \pm 20$ | $241 \pm 7$ | $52 \pm 3$ | $25 \pm 6$ | $105 \pm 10$ | $64 \pm 2$ |
| | QT-Opt+DrQ† | $102 \pm 5$ | $114 \pm 22$ | $243 \pm 5$ | $54 \pm 2$ | $26 \pm 5$ | $108 \pm 5$ | $65 \pm 1$ |
| | RSAC | $198 \pm 20$ | $257 \pm 89$ | $277 \pm 19$ | $244 \pm 17$ | $90 \pm 47$ | $112 \pm 17$ | $211 \pm 15$ |
| | CURL | $223 \pm 14$ | $136 \pm 24$ | $320 \pm 11$ | $170 \pm 10$ | $163 \pm 37$ | $222 \pm 37$ | $328 \pm 19$ |
| | CoRe | $\mathbf{480 \pm 23}$ | $\mathbf{706 \pm 39}$ | $\mathbf{354 \pm 26}$ | $\mathbf{354 \pm 10}$ | $\mathbf{540 \pm 73}$ | $\mathbf{445 \pm 48}$ | $\mathbf{479 \pm 31}$ |
| | CoRe (1M steps) | $684 \pm 24$ | $832 \pm 22$ | $483 \pm 29$ | $490 \pm 13$ | $810 \pm 60$ | $801 \pm 32$ | $686 \pm 42$ |

(d) Benchmark hard

| | Method | Mean | BiC-Catch | C-swingup | C-run | F-spin | R-easy | W-walk |
|---|---|---|---|---|---|---|---|---|
| Static | RSAC | $145 \pm 14$ | $113 \pm 12$ | $295 \pm 40$ | $140 \pm 6$ | $42 \pm 10$ | $74 \pm 3$ | $204 \pm 24$ |
| | CURL | $202 \pm 16$ | $163 \pm 36$ | $199 \pm 30$ | $239 \pm 12$ | $244 \pm 53$ | $121 \pm 16$ | $247 \pm 53$ |
| | CoRe | $\mathbf{499 \pm 28}$ | $\mathbf{710 \pm 37}$ | $\mathbf{447 \pm 10}$ | $\mathbf{339 \pm 13}$ | $\mathbf{809 \pm 14}$ | $\mathbf{197 \pm 21}$ | $\mathbf{490 \pm 15}$ |
| | CoRe (1M steps) | $638 \pm 27$ | $875 \pm 9$ | $554 \pm 11$ | $469 \pm 21$ | $898 \pm 12$ | $386 \pm 39$ | $645 \pm 29$ |
| Dyn | RSAC | $138 \pm 10$ | $165 \pm 19$ | $213 \pm 9$ | $151 \pm 10$ | $56 \pm 36$ | $86 \pm 4$ | $158 \pm 14$ |
| | CURL | $95 \pm 9$ | $103 \pm 18$ | $192 \pm 6$ | $78 \pm 13$ | $5 \pm 2$ | $73 \pm 13$ | $119 \pm 25$ |
| | CoRe | $\mathbf{307 \pm 22}$ | $\mathbf{436 \pm 48}$ | $\mathbf{257 \pm 18}$ | $\mathbf{200 \pm 11}$ | $\mathbf{364 \pm 83}$ | $\mathbf{234 \pm 61}$ | $\mathbf{353 \pm 25}$ |
| | CoRe (1M steps) | $467 \pm 29$ | $562 \pm 65$ | $350 \pm 26$ | $345 \pm 15$ | $620 \pm 97$ | $419 \pm 90$ | $505 \pm 17$ |

et al., 2021). However, a reasonable alternative could be to optimize $\Theta$ only using $J_M$ and keep the critic training separate. This would amount to separating the world model from the controller. As shown in Figure 9b, when excluding $\Theta$ from the critic vs including it, exclusion performs comparably on two tasks (ball_in_cup, cartpole), better on one (finger) and worse on three (cheetah, reacher, and walker). Overall, the inclusion regime works better. Therefore, we chose to include $\Theta$ in the

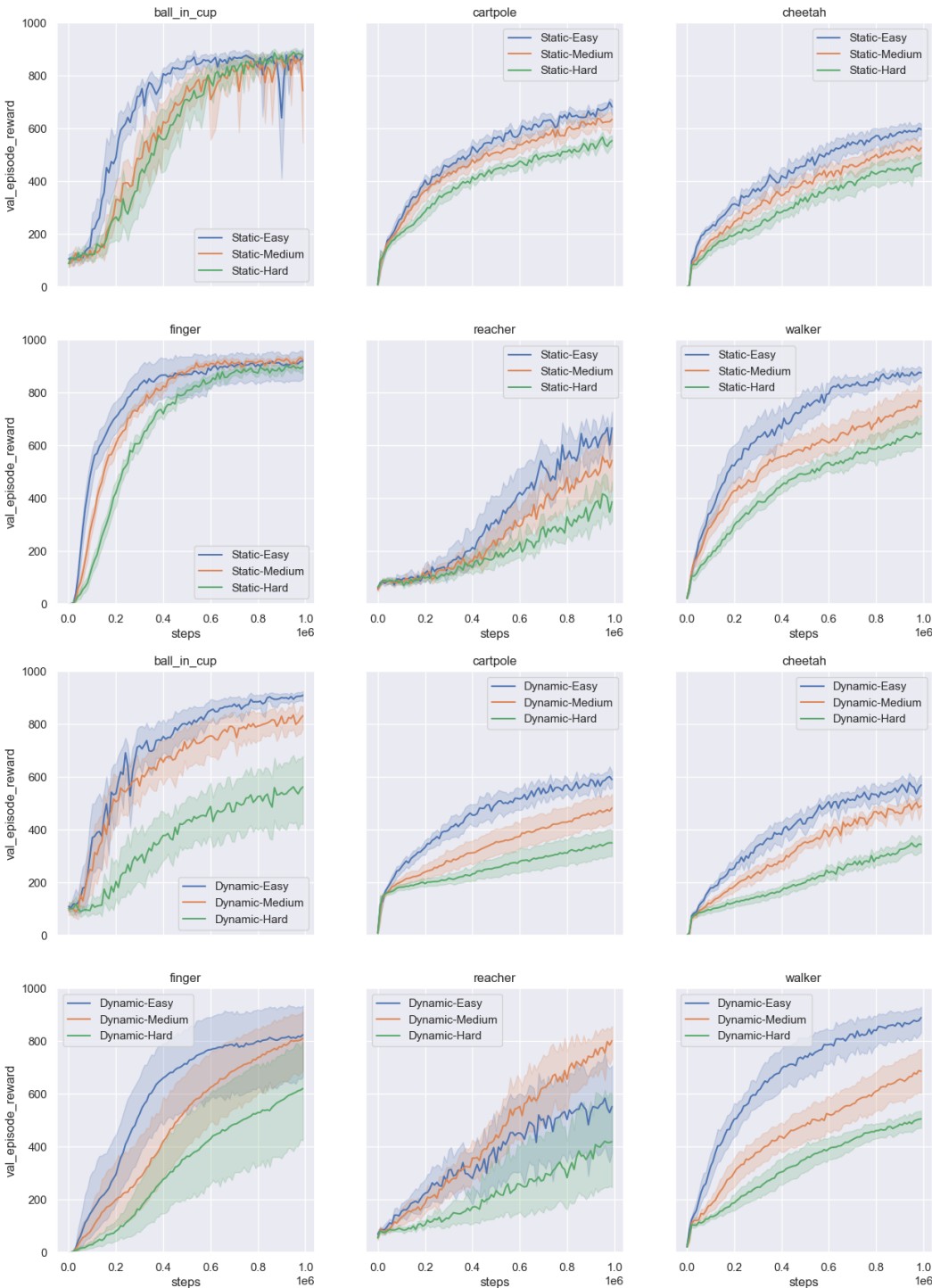

Figure 8: Progression of validation reward with training steps on all distraction settings in the Distracting Control Suite. Top two rows are for the static setting and bottom two for dynamic. Each plot shows easy, medium, and hard difficulty levels.

critic. In future work, an important direction to explore is the exclusion regime, especially in the multi-task setting, because separating the controller from the world model enables separating general understanding of the world from task-specific control policies, which is key to generalization.

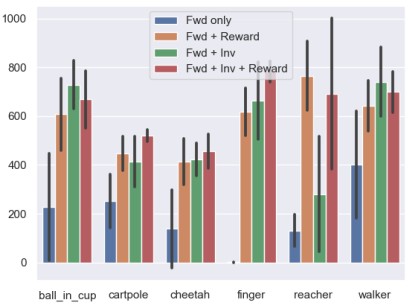
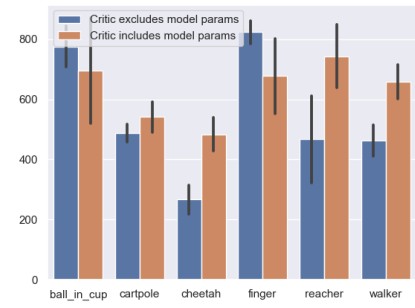

(a) Inverse dynamics and reward prediction.

(b) Optimizing $J_Q(\Theta, \theta_i)$ vs $J_Q(\theta_i)$

Figure 9: Additional ablation results. **Left** In addition to forward dynamics, inverse dynamics prediction and reward prediction improve performance. **Right:** Optimizing model parameters $\Theta$ using the critic loss is beneficial.

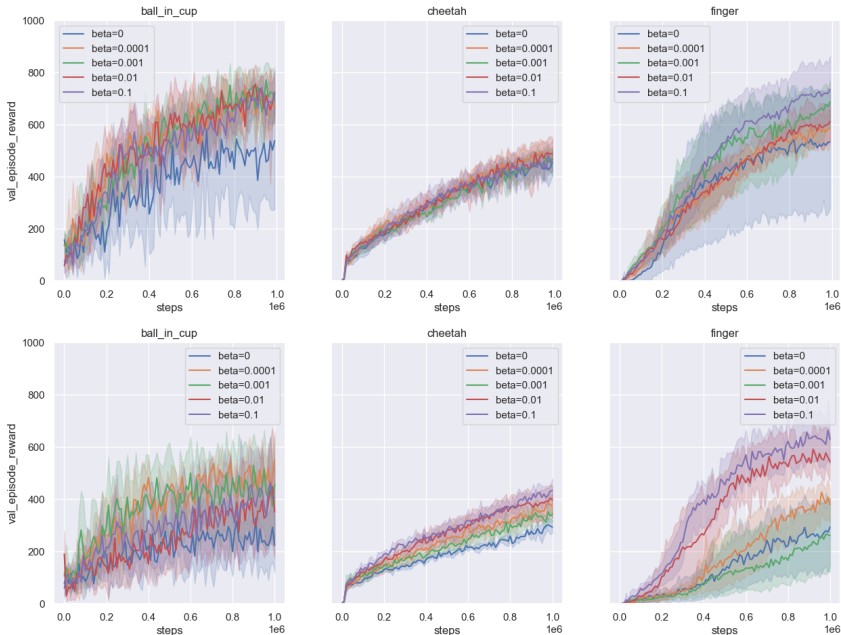

Figure 10: **Top:** Performance of CoRe for different values of the KL-loss weight $\beta$. **Bottom** Performance of a variant in which contrastive prediction is done from the posterior latent state, rather than the prior. We can see higher performance variance across values of $\beta$ when the posterior latent state is used.

### C.3 ROBUSTNESS TO $\beta$ WHEN RECONSTRUCTING FROM PRIOR VS POSTERIOR

CoRe predicts the next observation's feature from the *prior* latent state $\hat{z}_t$, making it different from sequential VAE-based models like SLAC (Lee et al., 2020) and Dreamer (Hafner et al., 2020) where the *posterior* latent state is used to reconstruct the observation. We argued in the paper that doing so makes the model more robust to the choice of $\beta$, the weight applied to the KL-term during optimization. In this experiment, we verify this argument by comparing CoRe with a variant where the posterior latent state is decoded to contrastively match the true observation's representation. We train both models with five values of $\beta$ and 5 different seeds each. In Figure 10 we can see that the posterior version (bottom row) has more variance in performance across different values of $\beta$. The proposed CoRe model (top row) is more stable, and hence, easier to train.

Figure 11: Results on the Robosuite Door Opening Task. All agents receive only RGB inputs.

## C.4 COMPARISON WITH MORE BASELINES FOR THE ROBOSUITE DOOR OPENING TASK

In this section, we present a comparison of CoRe with SAC+RAD, PSE, Recon, and CURL on the door opening task in Robosuite. The set of baselines is the same as that used for the Distracting Control Suite in Figure 2. Figure 11 shows the results. In this setting, the observations only come from the camera, i.e. proprioception is not provided separately. CoRe is able to perform significantly better than the baselines in all settings that involve distractions.

