# OpenReview forum: "Robust Robotic Control from Pixels using Contrastive Recurrent State-Space Models"
_ICLR.cc/2022/Conference — ICLR 2022 Submitted_

### Official Review · Reviewer_cJX3 · 2021-10-27

**Correctness:** 3
**Technical Novelty And Significance:** 1
**Empirical Novelty And Significance:** 2
**Recommendation:** 3
**Confidence:** 5

**Main Review:**

Weaknesses:

The major issue is that the proposed idea and the experiment setup is not novel. They highly overlap with a prior CoRL 2020 paper, Contrastive Variational Reinforcement Learning for Complex Observations (CVRL) [1], which, however, has not been cited in the submitted manuscript. They overlap in the following aspects:

1/ The idea is the same. CVRL also extends RSSM using contrastive learning and aims to improve the robustness of the learned model against real-world observations with high frequency noise. Both of them use InfoNCE for contrastive learning and the same RSSM structure.  In addition, both CVRL and the proposed method use the policy loss from dreamer for learning the policy network. The resulting equations and loss functions of the two algorithms are almost the same.

2/ CVRL also experimented on natural Mujoco games, which introduces moving backgrounds into the standard dm-control suites. This is exactly the same with the distracted dm-control suites used in the submitted manuscript.

Beyond what has been discussed in the paper, CVRL has mathematically shown that by replacing the generative observation likelihood with a contrastive objective, we can lower bound the original ELBO.

There are some other weaknesses, but I believe the issues discussed above are sufficient to make it a clear rejection.

**Summary Of The Paper:**

This paper has presented CoRe, Contrastive Recurrent state space model, for model-based robust model-based reinforcement learning for robotic control. Standard reconstruction-based state space models are less robust in the unstructured real-world scenarios because of the high-frequency details. Instead, CoRe learns the state space model with contrastive learning, which greatly improves robustness. In addition to this, a policy is being learned with SAC. Experiments on distracting control suites and several robotic control tasks demonstrate the better robustness of CoRe.

**Summary Of The Review:**

The paper is highly similar to a prior work as mentioned above, and as a result, the contribution of the paper is very limited. I would vote for a rejection.

---

> ### Author Response · Authors · 2021-11-22
> **Addressing novelty with respect to CVRL**
>
> We thank the reviewer for pointing out an important related work. We address the concerns raised by the reviewer below-
>
> > 1/ The idea is the same. CVRL also extends RSSM using contrastive learning and aims to improve the robustness of the learned model against real-world observations with high frequency noise. Both of them use InfoNCE for contrastive learning and the same RSSM structure. In addition, both CVRL and the proposed method use the policy loss from dreamer for learning the policy network. The resulting equations and loss functions of the two algorithms are almost the same.
>
> The method proposed in CVRL is indeed similar to CoRe. However, there is a crucial difference in the objective function. In CVRL, mutual information maximization is between the observation and the *posterior* latent state (i.e. in Eq 6 in the CVRL paper, the contrastive loss term uses an expectation over the posterior distribution $q$ and the latent state $s_t$ is computed using observations up to and including time $t$). On the other hand, CoRe models the observation using the *prior* latent state. In other words, the latent state used in the contrastive loss in CoRe has not yet seen the observation $o_t$, but the one in CVRL has. We found that using the prior to model the observation improves not only the overall performance but also stability w.r.t the weighting of the KL divergence term with respect to the contrastive loss term. This is discussed in our paper in Section 2.2 in the paragraph titled “Relationship with Action-Conditioned Sequential VAEs” and is studied in an ablation experiment in Appendix C.3. The intuition is that in sequential VAE-like models (e.g CVRL and Dreamer) only the KL term does prediction. The other term, which is the contrastive loss (in CVRL) or the likelihood loss (in Dreamer) models the data. In CoRe, on the other hand, the contrastive term also involves future prediction because it must model the current observation given the previous state and action. This helps alleviate the posterior collapse problem and makes CoRe less reliant on careful tuning of the beta parameter that weights the KL term term relative to the contrastive loss. Posterior collapse is a well known problem with beta-VAE like models (see https://github.com/sajadn/posterior-collapse-list) which CoRe can address. Therefore, we believe that the difference in the objective function w.r.t CVRL is significant.
>
> > 2/ CVRL also experimented on natural Mujoco games, which introduces moving backgrounds into the standard dm-control suites. This is exactly the same with the distracted dm-control suites used in the submitted manuscript.
>
> In our paper, we use not just moving backgrounds but a moving camera and changing object colors simultaneously to increase the severity of the distractions. To the best of our knowledge, this is the first paper to solve pixel-based control tasks in this extremely hard setting. Previous works (e.g. CVRL, DBC, PSE) only attempt to solve the tasks with moving backgrounds, keeping the color and position of the robot agent (relative to the camera) fixed. Therefore, our experimental setting is significantly more challenging that previous works. The authors of the Distracting Control Suite benchmark (which we use in our experiments) mention that different types of distractions happening simultaneously has a very strong negative effect on the performance of baseline models. The fact that our model can solve these very hard distraction settings demonstrates a non-trivial extension in the capability of this class of models.

---

> > ### Comment · Reviewer_cJX3 · 2021-11-27
> > **[Reviewer Response] Not ready for publication yet.**
> >
> > I thank the authors for their detailed feedback. It helped to clarify some of my concerns and I agree that CoRe and CVRL are different in terms of: (1) CoRe introduces the prior based prediction, while CVRL might suffer from posterior collapse; (2) The experiment setup is more challenging in CoRe. These contributions are indeed valuable to the community.
> >
> > However, as Reviewer skrV has also discussed, major revisions are necessary for the current manuscript. CVRL and CPC|A (as discussed by Reviewer skrV), to some extent, weakens the novelty of the work. CoRe improves the existing methods from different perspectives, but the original claim of the paper might not be well supported by the current presentation and experiment. For the presentation, it is important to redo a thorough literature review and compare the differences with existing works. Currently, lots of important literature is missing. As the idea of using contrastive learning is no longer new for RSSM, the prior-based prediction should be the focus instead. In terms of the experiments, although the current setup is more challenging than the previous ones, there is no evidence that CVRL / CPC|A would fail. Additional experiments should be done.
> >
> > In summary, I believe that the work has made important contributions, but its current presentation and experiments are not ready for publication yet. I’ve increased my recommendation score, but keep it a rejection for now.

---

### Official Review · Reviewer_skrV · 2021-11-03

**Correctness:** 3
**Technical Novelty And Significance:** 2
**Empirical Novelty And Significance:** 3
**Recommendation:** 5
**Confidence:** 4

**Main Review:**

# Strengths
* The clarity of the paper is good and the approach has been described well.
* The experiments have been well designed and indicate strong improvements in the model robustness (particularly Figure 2 and Table 3). Table 2 was helpful to understand the key design choices and their impact on final performance.
* The mask visualizations in Figure 5 provide good clarity on how the robustness is achieved by the proposed method.
* The ablation studies in the main paper and supplementary clearly highlight the benefits of individual components of the method.


# Weaknesses

## Novelty
* The novelty of the approach isn't quite clear. In Sec. 1 (page 2), the authors highlight that one of the key findings is that "contrastive learning can in fact lead to surprisingly strong robustness to severe distractions, provided that a recurrent state-space model is used". This finding itself is valuable in my opinion, but the method "CoRe" does not seem novel to me. Relative to Dreamer and SLAC, the key novelty appears to be the contrastive loss term from Eqn (1) which predicts future observations instead of auto-encoding. However, the idea of using a recurrent model to predict encodings of future observations (i.e., not an auto-encoder) has been studied in CPC [1], action-conditioned CPC (aka CPC|A) [2], Predictions of Bootstrapped Latents (aka PBL) [3], etc. CPC does not use action conditioning, but uses contrastive learning. CPC|A uses action conditioning + contrastive learning (very similar to CoRE). PBL uses action conditioning + reconstruction (similar to "Recon" baseline, but has an additional loss to predict state conditioned on observation representation).
* Given the above, the methodological novelty of CoRe is not clear to me (particularly, relative to CPC|A). Is there something different about the contrastive loss in CoRe (relative to CPC|A) without which the performance degrades severely? Or is the novelty only in the observation that recurrent models are needed for contrastive loss to work well? Note, while some recent work like DBC does not use recurrent models, prior approaches already use them with contrastive loss.


## Intuition behind why recurrent state models are needed for contrastive learning
*  [End of section 1] The authors suggest that when recurrent models are used along with contrastive learning, the smoothness of the state-space ensures the presence of informative hard negatives in the same mini-batch of training. This isn't clear to me. Positives and negatives are obtained from the next-step real observation's encodings (just a single frame) and does not use the recurrent model. How is the smoothness of the state-space related to hard negatives? While the performance degrades without the recurrent model in Figure 3 (top), isn't this more likely due to a poorer state representation caused by the lower capacity (no RNN) and lack of information aggregation over time? This would affect all models, not just the CoRe.


## Missing baselines / related works
A few related methods have not been compared with or discussed.

* Augmented Temporal Contrast (aka ATC) [4] does not use recurrence, but uses future observation prediction and has been shown to achieve good improvements over CURL.
* CPC|A [2] uses recurrence, future observation prediction (contrastive), and action conditioning. This is very similar to the proposed method and should be compared with.
* PBL [3] uses recurrence, future observation prediction (reconstruction), and action conditioning. This method introduces an additional P(state | observation) term that results in bootstrapped learning (might improve over the Recon baseline).


## Other concerns

* Why is CoRe worse than PSE on 4 / 6 tasks for DAVIS (2 videos) in Table 1? Why is CoRe better with 60 videos?
* Why are only 2 baselines used in Figure 4? Can the authors include the complete set? Do we observe similar trends as Figure 2?
* Are the findings from Figure 5 specific to CoRe? Or do other methods (like PSE) also learn similar masking functions?



[1] CPC - Oord, Aaron van den, Yazhe Li, and Oriol Vinyals. "Representation learning with contrastive predictive coding." arXiv preprint arXiv:1807.03748 (2018).
[2] CPC|A - Guo, Zhaohan Daniel, et al. "Neural predictive belief representations." arXiv preprint arXiv:1811.06407 (2018).
[3] PBL -  Guo, Zhaohan Daniel, et al. "Bootstrap latent-predictive representations for multitask reinforcement learning." International Conference on Machine Learning. PMLR, 2020.
[4] ATC - Stooke, Adam, et al. "Decoupling representation learning from reinforcement learning." International Conference on Machine  Learning. PMLR, 2021.

**Summary Of The Paper:**

The paper proposes a recurrent state-space model that learns robust representations for robotic control. The proposed method builds on top of prior works on world-models which learn a latent dynamics model of the agent which can be used for planning and action selection. Different from prior work such as Dreamer and SLAC which rely on pixel-based observation reconstruction, this paper highlights that a simpler contrastive loss for the next-observation prediction achieves better results **if** a recurrent state-space model is used for the latent space. Results are presented on the Distracting Control Suite benchmark and show strong improvements over prior approaches.

**Summary Of The Review:**

I am concerned about the lack of clear novelty in the paper, and other experimental issues highlighted in "Weaknesses". I will update my rating based on the author's responses.

---

> ### Author Response · Authors · 2021-11-22
> **Response to other concerns**
>
> >Why is CoRe worse than PSE on 4 / 6 tasks for DAVIS (2 videos) in Table 1? Why is CoRe better with 60 videos?
>
> CoRe performs worse than PSE with 2 training videos due to overfitting. For example, on the reacher task, CoRe gets an average validation reward of 610 but the training reward (not reported in the table) is 976, which is better than the reward that PSE gets (955). Since training with only 2 background videos is a fairly extreme setting, we did not spend a lot of time tuning our model for this case. With 60 videos, overfitting is less of a concern. All of the hyper-parameters, including model size, were tuned in the 60 video setting. CoRe results are likely to improve for the 2 video setting if more careful regularization and model size tuning are done.
>
> >Why are only 2 baselines used in Figure 4? Can the authors include the complete set? Do we observe similar trends as Figure 2?
>
> We included the 2 baselines RCURL and RSAC because they correspond to ablating the important attributes of the model. In the updated version of the paper, we added the other baselines for the Robosuite environment (Appendix C.4) and observe similar trends.
>
> > Are the findings from Figure 5 specific to CoRe? Or do other methods (like PSE) also learn similar masking functions?
>
> We experimented with masking encoders for SAC+RAD and PSE in the medium dynamic distraction setting but found that those models were neither able to get good performance nor learn interpretable masks.

---

> ### Author Response · Authors · 2021-11-22
> **Addressing concerns on novelty and importance of recurrence in contrastive learning**
>
> We thank the reviewer for raising thoughtful questions and pointing out relevant prior work.
>
> ## Novelty and relationship to other methods
>
> >Augmented Temporal Contrast (aka ATC) [4] does not use recurrence, but uses future observation prediction and has been shown to achieve good improvements over CURL.
>
> We cite ATC in the updated version of the paper. Similar to CoRe, ATC does future prediction in a contrastively-learned space. However unlike CoRe, it does not use action-conditioned predictions, recurrence, or sequential correction of the latent state. It only operates on the observation space (similar to CURL), completely separated from the RL objective.
>
> > CPC|A [2] uses recurrence, future observation prediction (contrastive), and action conditioning. This is very similar to the proposed method and should be compared with.
>
> CoRe is different from CPC|A because besides the contrastive loss, the CoRe objective also includes a KL term that corrects the predicted latent state using the current observation. CoRe would reduce to CPC|A if we take out the "Corrector" component shown in Figure 1. Figure 10 (Top row) in the appendix shows results when setting $\beta_\text{KL} =0$ which corresponds to CPC|A.
>
> >PBL [3] uses recurrence, future observation prediction (reconstruction), and action conditioning. This method introduces an additional P(state | observation) term that results in bootstrapped learning (might improve over the Recon baseline).
>
> We cite PBL in the updated version of the paper. Unlike CoRe, PBL uses future prediction in the observation space. Based on our experiments with the Recon baseline we can already conclude that reconstruction in the pixel space negatively impacts performance when significant distractions are present, which is the main point we wanted to make regarding reconstruction-based methods.
>
> Please see response to reviewer cJX3 for differences with respect to CVRL.
>
> ## Role of recurrence in contrastive learning
>
> > Positives and negatives are obtained from the next-step real observation's encodings (just a single frame) and does not use the recurrent model. How is the smoothness of the state-space related to hard negatives?
>
> Our model uses a symmetric contrastive loss (Eq 1) which consists of two terms. In the first term, the anchor is the prediction from the recurrent state and the positives and negatives come from the observations (as the reviewer pointed out). In the second term, the anchor is the real observation encoding and the positives and negatives come from the recurrent states. For the second term, the smoothness of the state-space directly provides hard negatives. For the first term, the smoothness of the recurrent state contributes to the hardness of task in the following way - suppose $(y_t, x_t, x_{t+1})$ and $(y_{t+1}, x_{t+1}, x_{t})$ were triplets of the form (anchor, positive, negative). Then the loss is asking for $y_t$ and $x_t$ to be similar (from the first triplet) and $y_{t+1}$ and $x_{t}$ to be dissimilar (from the second triplet). So the objective still has to distinguish between consecutive outputs from the RNN, $y_t$ and $y_{t+1}$, using $x_t$.
>
> > While the performance degrades without the recurrent model in Figure 3 (top), isn't this more likely due to a poorer state representation caused by the lower capacity (no RNN) and lack of information aggregation over time?
>
> We created high-capacity feed-forward models that take stacked frames as input. Since the tasks don’t really require long-term context, we think that stacking a few frames and using a feed-forward model with a large number of parameters is just as powerful as an RNN in terms of representation power. We compare to other recurrent baselines (RSAC and RCURL) which use the same recurrent architecture as CoRe. Those models don’t perform as well, which shows that the recurrent architecture by itself is not sufficient to solve the task.

---

> > ### Comment · Reviewer_skrV · 2021-11-25
> > **Reviewer response: Significant changes may be needed in write-up (regarding novelty) and experiments (more baselines).**
> >
> > I thank the authors for their detailed feedback. A good part of my concerns were addressed. But the following concerns still remain.
> >
> >
> > ## Novelty concern
> >
> > "contrastive learning can in fact lead to surprisingly strong robustness to severe distractions, provided that a recurrent state-space model is used" --- this is the key novel finding in the paper. However, as suggested by R-cJX3 and myself, this combination of contrastive learning with the recurrent state-space model is not new (other works which do this include CPC|A and CVRL). The authors do carefully distinguish from these methods in their response, but looking at the bigger picture, these are technical changes which have little to do with their claimed novelty (of showing contrastive learning + RSSM provides strong robustness). If we were to extend Table 2 to accommodate these papers, 2 more rows would be needed.
> >
> > * Row 4 - prior-based prediction --- this states whether the contrastive loss, or more generally, the comparison b/w inferred state and observation is prior-based (i.e., state is inferred purely from past state and action), or posterior-based (i.e., state is inferred from past state, action, and current observation).
> > * Row 5 - KL-div loss --- this states whether the KL-divergence loss from page 3 is included in the total loss or not.
> >
> > CVRL would have the features from rows 1-3 and 5. CPC|A would have the features from 1-4. CoRe would have all 5 features. In effect, CoRe would be a careful combination of prior work, which is definitely valuable to have. However, the current novelty claims do take a hit, and significant changes have to be made in the writing.
> >
> > **Summary:** The fact that CoRe's performance may be better (still not sure if this is true, see next point) due to prior-based prediction and KL-div loss  does not justify the current novelty claims.
> >
> > ## Experimental concerns
> >
> > ### Using the complete CVRL and CPC|A as baselines.
> >
> > The authors have carefully distinguished from CVRL and CPC|A. But there are other aspects of the two methods which have not been considered.
> >
> > * CVRL includes an MPC component which was shown to be important to their performance (Table 2 from CVRL).
> > * CPC|A does not include the KL term, but includes longer horizon prediction.
> >
> > I'm left wondering how CoRe compares with the complete CVRL and CPC|A models, and whether these additional features used in these methods somehow compensate for the lack of prior-based prediction and KL-div loss, respectively.
> >
> > ### Careful comparison with PBL
> > > Unlike CoRe, PBL uses future prediction in the observation space. Based on our experiments with the Recon baseline we can already conclude that reconstruction in the pixel space negatively impacts performance when significant distractions are present, which is the main point we wanted to make regarding reconstruction-based methods.
> >
> > * PBL does not perform pixel-space observation reconstruction, which was one of their key features. A "latent encoding" of the observation is reconstructed. Therefore, a 'Recon-latent' baseline with latent observation reconstruction is missing. For some reason, I misunderstood the current Recon baseline to be predicting the latent features rather than pixels. This simple extension of Recon is critical to understand whether the contrastive loss is necessary or not.
> >
> > * In addition to the future latent encoding prediction, the PBL authors also predict the state from the latent observation encoding. This also seems important for robustness since the "latent encoding" can learn to capture only parts of the observation space that are important for the task (i.e., only the foreground) and ignore the rest (the background).
> >
> > In short, the both the 'Recon-latent' baseline (where only the latent features are predicted), and the full PBL model need to be compared with to convincingly say that the contrastive prediction is better.
> >
> > **Summary:** Stronger baselines are missing and it is unclear to me if CoRe is the best performing method.
> >
> > I feel that this paper cannot be accepted without non-trivial changes to both the writing and experiments.

---

### Official Review · Reviewer_FBRd · 2021-11-03

**Correctness:** 4
**Technical Novelty And Significance:** 3
**Empirical Novelty And Significance:** 3
**Recommendation:** 5
**Confidence:** 3

**Main Review:**

## Strong points

The paper is well written and easy to follow. I particularly
liked the more straight-to-the-point and honest approach on writing.
The authors could have easily try to mask their similarity with some
other methods in the literature but they were upfront with that.
This made the understanding of the paper way easier and also
easier to understand the main contributions.

There is a substantial comparison with relevant models in the literature that already
deal with the DCS. The results obtained are really impressive
and are a new state-of-the-art  on interactive environments
with distractive backgrounds.

The ablations specifically the ones concerning the usability of the recurrent
contrastive method are very useful. Specially the intuition on the negative
samples per mini batch.
Further, the appendix provides even more ablations, making very
clear that a recurrent state has a major impact on the general performance.


The visualizations are also good standard to be used on this field which performs
control directly from pixels. They also show that their model is capable of directly
detecting the object of interest.

##  Weak points

The proposed model is indeed very similar to Dreamer[1]. The authors address specifically
one point from dreamer: the contrastive learning strategy, which failed to produce better
results as reported by [1]. However, I do believe that this closer look into this
specific point can be useful for the community and insightful in general.

## Questions

I would be curious to find more references on the idea that hard negatives needs to be found
on the mini batch in order to contrastive learning to have a good performance.

I wonder how the performance of this method would be in a more open task than mujoco
style robotic control. Tasks with a different initial state or with other elements in the scene
that the agent need to interact might have a negative impact on this method results.


[1] Hafner, Danijar, et al. "Dream to control: Learning behaviors by latent imagination." arXiv preprint arXiv:1912.01603 (2019).

**Summary Of The Paper:**

The paper shows a latent-planning RL model  to
learn control policies directly from pixels. To learn a better
representation it uses recurrent model contrastive learning approach
which enhances the representation learning performance of single
frame based contrastive methods.
This was tested robotic control suites with challenging distracting
backgrounds.

The main contribution of the work is the addition of recurrence to the
model and the extensive testing and explanation on the reason it tends
to work better.

**Summary Of The Review:**

I think this is a very useful and well written paper.
Even though the scope is small, the results are convincing
and it shows a very clear way on how to effectively use contrastive
representation learning methods  while learning to control directly from
pixels.


## After Rebuttal

After reading the other reviewers comments and the rebuttal
I see that i missed some literature that needed further comparison.
I think it is a good and well written paper but I would lean for rejection
given this new data.

---

> ### Author Response · Authors · 2021-11-22
> **Response**
>
> > The results obtained are really impressive and are a new state-of-the-art on interactive environments with distractive backgrounds.
>
> We would like to thank the reviewer for recognizing the quality of the results.
>
> > I would be curious to find more references on the idea that hard negatives needs to be found on the mini batch in order to contrastive learning to have a good performance.
>
> The SimCLR paper (https://arxiv.org/abs/2002.05709) Section 5.2 empirically validates that “In contrast to supervised learning, in contrastive learning, larger batch sizes provide more negative examples, facilitating convergence”. Another good reference is the SwaV paper (https://arxiv.org/abs/2006.09882) which tries to address the problem of needing a large number of explicit pairwise feature comparisons by learning prototypes.
>
> >I wonder how the performance of this method would be in a more open task than mujoco style robotic control. Tasks with a different initial state or with other elements in the scene that the agent need to interact might have a negative impact on this method results.
>
> We thank the reviewer for suggesting these environments. In future work, we plan to address the challenges brought on by more complex environments.

---

### Decision · Program_Chairs · 2022-01-20

**Decision:**

Reject

**Comment:**

Meta Review of Robust Robotic Control from Pixels using Contrastive Recurrent State-Space Models

This work investigates a recurrent latent space planning model for robotic control from pixels, but unlike some previous work such as Dreamer and RNN+VAE-based World Models, they use a simpler contrastive loss for next-observation prediction. They presented results on the DM-control suite (from pixels) with distracting background settings. All reviewers (including myself) agree that this is a well-written paper, with clear explanation of their approach. The main weaknesses of the approach are on the experimental side (see review responses to author’s rebuttal by skrV and cjX3). Another recommendation from me is to strengthen the related work section to clearly position the work to previous work - there is clear novelty in this work, but this should be done to avoid confusion. The positive sign is that in the discussion phase, even the very critical cjX3, had increased their score and acknowledged the novelty from previous related work. In the current state, I cannot recommend acceptance, but I’m confident that with more compelling experiments recommended by the reviewers, and better positioning of the paper to previous work, I believe that this paper will surely be accepted at a future ML conference or journal. I’m looking forward to seeing a revised version of this paper for publication in the future.